# Associations between Training Load and Well-Being in Elite Beach Soccer Players: A Case Report

**DOI:** 10.3390/ijerph19106209

**Published:** 2022-05-20

**Authors:** Júlio A. Costa, Pedro Figueiredo, Alberto Prata, Tiago Reis, Joana F. Reis, Luís Nascimento, João Brito

**Affiliations:** 1Portugal Football School, Portuguese Football Federation, FPF, 1495-433 Cruz Quebrada, Portugal; pedfig@me.com (P.F.); alberto.prata@fpf.pt (A.P.); tiago.reis@fpf.pt (T.R.); joana.reis@fpf.pt (J.F.R.); luis.nascimento@physioclem.pt (L.N.); joao.brito@fpf.pt (J.B.); 2CIDEFES, Universidade Lusófona, Campo Grande, 1749-024 Lisboa, Portugal; 3Research Center in Sports Sciences, Health Sciences and Human Development (CIDESD), 5001-801 Vila Real, Portugal; 4Laboratory of Physiology and Biochemistry of Exercise, Faculdade de Motricidade Humana, Universidade de Lisboa, 1495-751 Lisboa, Portugal; 5Interdisciplinary Center for the Study of Human Performance (CIPER), Faculdade de Motricidade Humana, Universidade de Lisboa, 1495-751 Lisboa, Portugal; 6Polytechnic Institute of Leiria, 2411-901 Leiria, Portugal

**Keywords:** training monitoring, performance, football, team sports

## Abstract

The current case study aimed to quantify within-subjects correlations between training load and well-being in elite male beach soccer players. Data were obtained over three consecutive days during the preparation camp for the FIFA Beach Soccer World Cup Russia 2021. The session rating of perceived exertion (s-RPE) and external training load metrics using global positioning system (GPS) were recorded. Eleven players reported perceived well-being (sleep quality, fatigue, muscle soreness and stress) using a Likert scale (Hooper Index) before breakfast. Within-subjects correlation coefficients between workload and well-being were calculated. Workload metrics and perceived well-being indices were significantly lower on day three than on days one and two. The Hooper Index presented a *very large* positive correlation with s-RPE (*r* = 0.86 [0.67, 0.94], 95% confidence interval, CI), exposure time (*r* = 0.88 [0.71, 0.95]), total distance (*r* = 0.83 [0.60, 0.93]), high-speed distance (*r* = 0.77 [0.50, 0.91]), and number of sprints (*r* = 0.75 [0.47, 0.90]). Sleep quality presented a *moderate* to *large* positive correlation with s-RPE (*r* = 0.51 [0.11, 0.77]), exposure time (*r* = 0.50 [0.10, 0.76]), high-speed distance (*r* = 0.53 [0.15, 0.78]), number of sprints (*r* = 0.62 [0.28, 0.83]) and total distance (*r* = 0.41 [0.18, 0.78]). Fatigue presented a *large* to *very large* positive correlation with s-RPE (*r* = 0.85 [0.66, 0.94]), exposure time (*r* = 0.90 [0.78, 0.96]), total distance (*r* = 0.86 [0.68, 0.94]), high-speed distance (*r* = 0.65 [0.31, 0.84]) and number of sprints (*r* = 0.56 [0.18, 0.79]). Muscle soreness presented a *large* to *very large* positive correlation with s-RPE (*r* = 0.79 [0.56, 0.91]), exposure time (*r* = 0.83 [0.62, 0.93]), total distance (*r* = 0.81 [0.59, 0.92]), high-speed distance (*r* = 0.75 [0.47, 0.89]) and number of sprints (*r* = 0.59 [0.22, 0.81]). Overall, workload presented a meaningful correlation with perceived well-being indices in elite male beach soccer players during a training camp. These findings suggest that workload metrics and perceived well-being indices can be implemented into the daily routine of an elite beach soccer team, which may assist coaches, sports scientists, and practitioners in better preparing players for beach soccer competitions.

## 1. Introduction

Beach soccer is one of the official football codes ruled by FIFA. In its official form, beach soccer is a 5-a-side game played on a 35–37 m long and 26–28 m wide pitch covered with sand; it is played worldwide by amateur and professional players [1]. As recently described [2], beach soccer is characterized as an intermittent high-intensity sport involving specific actions such as accelerations, jumps, and passes with the added difficulty of executing these skills on an unstable surface (i.e., sand).

As in any elite sport, daily monitoring of a player’s internal and external training loads is critical in beach soccer because high training loads coupled with inadequate recovery can result in injury, illness, or overtraining [3]. For that purpose, a commonly used non-invasive method of monitoring perceived training load is the session rating of perceived exertion training load (s-RPE training load: session duration (in minutes) × RPE (using either CR-10, CR-100 or 6–20 scales)) [4]. The s-RPE has been extensively used alongside global positioning system (GPS)-derived variables of training load in other football codes (e.g., soccer) to monitor changes in individual training or match performance [5]. In addition, monitoring external loads via GPS and accelerometers, together with subjective ratings of well-being or mood states before each training session may provide information about a player’s perceived response to the global training load in team sports [6].

For the development of training processes in beach soccer players, the careful monitorization of training load seems to be relevant [2]. Furthermore, a suitable training load is fundamental for short-term performance development and to empower beach soccer players [2]. In order to complement the training load monitorization and to avoid unbalance in athletic performance, special consideration should also be applied to well-being variables [7]. More specifically, the Hooper-Index and/or its subsets (i.e., sleep quality and the quantities of stress, fatigue, and delayed onset muscle soreness [DOMS]) [8] have been shown as promising tools for monitoring fatigue in soccer players [9,10,11]. The Hooper-Index, in particular, has been reported to be associated with training load in professional soccer players [12]. In addition, intense training may also have an influence on the physical factors that affect athletic performance, such as sleep and recovery quality, stress, fatigue, and muscle soreness (i.e., the Hooper-Index and/or its subsets) [13]. Thus, the overall goal of training monitoring is to identify the biological and physical effects that training sessions and/or matches have on players [14]. Moreover, the Hooper index is defined as a psychological questionnaire for monitoring changes in training-related stress, wellness, strain and recovery, and have been suggested to allow the detection of early signs of tiredness and/or overtraining in high-performance sport programs [13].

In this sense, the relationship between well-being (assessed based on a self-report questionnaire relative to sleep quality, DOMS, fatigue, and stress [8]) and workload metrics (e.g., s-RPE, total distance, high-speed distance and number of sprints [1]) has received growing interest in recent years [7]. Moreover, the literature provides significant interactions between DOMS, stress, fatigue perception, and sleep quality with training load metrics [7].

However, to date no studies have analyzed the association between workload metrics and perceived well-being indices in elite male beach soccer players. Therefore, this case study aims to quantify the within-subjects correlations between workload metrics and perceived well-being indices in elite male beach soccer players during three consecutive days of a preparation camp for the FIFA Beach Soccer World Cup Russia 2021.

## 2. Materials and Methods

### 2.1. Participants

Eleven elite male soccer players (aged: 29.4 ± 6.9 years; height: 1.82 ± 0.06 m; body mass: 74.1 ± 7.9 kg (mean ± SD)) from the Portuguese national team participated in the study. The study was approved by the Ethics Committee of the Portugal Football School (CE PFS 6/2021). The players had to have participated in all training sessions during the training camp to be included in the analysis. Goalkeepers (*n* = 3) were not included in the analyses, and one player decided not to use the GPS device during training sessions.

### 2.2. Procedures

A longitudinal observational case study design was adopted. Data were collected for three consecutive days of training, during the preparation camp for the FIFA Beach Soccer World Cup Russia 2021. The players were hosted in the same hotel (Nazaré, Portugal). The players slept in shared twin rooms with separate beds (allocated by the technical staff). The daily schedule programs were: time to wake up until 9:00, breakfast until 9:30, lunch at 13:00, dinner at 20:00 and return to rooms at 22:00. All training sessions (starting at 16:00) were held on a beach soccer pitch near the hotel (~2 min walking).

Players reported individual RPE using the Borg category ratio scale (CR10) after each training session, which was the usual routine of the investigated male beach soccer players. The CR10 score (perceived intensity) was subsequently multiplied by the individual exposure time (training volume), and thus provided an overall load quantification of the session (i.e., s-RPE) [4].

Players used 10-Hz GPS units during training sessions (STATSports Apex, Northern Ireland) [15]. External load variables included total training duration (i.e., exposure time), total distance covered, high-speed distance (>13 km/h^−1^) [1] and the number of sprints (>18 km/h^−1^) [1].

In addition, players reported individual well-being using the Hooper Index [8] on the morning of each training day (i.e., prior to breakfast). The Hooper Index uses a Likert scale and collects well-being ratings relative to sleep quality, fatigue, muscle soreness and stress (scale for sleep quality 1 [very, very good]–7 [very, very bad]) and scale for fatigue, muscle soreness and stress 1 [very, very low]–7 [very, very high]); The Hooper Index is the summation of these four ratings.

### 2.3. Statistical Methods

Sample distribution was tested using the Shapiro–Wilk test for workload metrics and perceived well-being indices during the training camp. Differences in workload metrics and perceived well-being indices between training days were examined using linear mixed model (Lmm) analysis. In addition, the assumption for normally distributed residuals was graphically tested (Q-Q plot) and no deviations from normal distribution were visible in the Lmm analyses. The level of significance for statistical comparisons was set at 0.05. The days with training sessions were included as a fixed effect and player identity (subject ID) as the random effect. The variance-covariance structures were selected according to the smallest Akaike Information Criterion. Bonferroni pairwise comparisons were used to show the differences between days of training for workload metrics and perceived well-being indices.

Within-subjects correlations (*r*, 95% confidence interval, CI) [16] were tested between workload metrics (i.e., s-RPE, exposure time, total distance, high-speed distance and sprints) and perceived well-being indices (i.e., sleep quality, fatigue, stress and muscle soreness; Hooper index) across three days of data collection. We qualitatively interpreted the magnitudes of correlation using the following criteria: *trivial* (*r* ≤ 0.1), *small* (*r* = 0.1–0.3), *moderate* (*r* = 0.3–0.5), *large* (*r* = 0.5–0.7), *very large* (*r* = 0.7–0.9) and *almost perfect* (*r* ≥ 0.9) [17]. When the 95% CI overlapped positive and negative values, the effect was deemed to be *unclear*.

Lmm statistical analyses were conducted using SPSS software (version 27.0.1, SPSS Inc., Chicago, IL, USA). For within-subjects correlations, the rmcorr [18] package was used in R statistical software (version 3.4.1, R Foundation for Statistical Computing, Vienna, Austria).

## 3. Results

Group means and 95% CI estimates in workload metrics and group median and interquartile ranges for perceived well-being indices are presented in Table 1. Workload metrics (i.e., s-RPE, exposure time, total distance, high-speed distance and number of sprints) and perceived well-being indices (i.e., sleep quality, fatigue, muscle soreness and stress; Hooper Index) were significantly lower on day three compared with day one and day two.

The within-subject correlations between workload metrics and perceived well-being indices are presented in Figure 1.

The Hooper Index presented *very large* positive correlations with s-RPE (*r* = 0.86 [0.67, 0.94]; *p* < 0.001), exposure time (*r* = 0.88 [0.71, 0.95]; *p* < 0.001), total distance (*r* = 0.83 [0.60, 0.93]; *p* < 0.001), high-speed distance (*r* = 0.77 [0.50, 0.91]; *p* < 0.001), and number of sprints (*r* = 0.75 [0.47, 0.90]; *p* < 0.001).

Sleep quality presented *large* positive correlations with s-RPE (*r* = 0.51 [0.11, 0.77]; *p* = 0.01), exposure time (*r* = 0.50 [0.10, 0.76]; *p* = 0.01), high-speed distance (*r* = 0.53 [0.15, 0.78]; *p* = 0.001) and number of sprints (*r* = 0.62 [0.28, 0.83]; *p* = 0.001). A *moderate* positive correlation was found between sleep quality and total distance (*r* = 0.41 [0.18, 0.78]; *p* = 0.03).

Fatigue presented *very large* positive correlations with s-RPE (*r* = 0.85 [0.66, 0.94]; *p* < 0.001), exposure time (*r* = 0.90 [0.78, 0.96]; *p* < 0.001) and total distance (*r* = 0.86 [0.68, 0.94]; *p* < 0.001). Fatigue also showed *large* positive correlations with high-speed distance (*r* = 0.65 [0.31, 0.84]; *p* = 0.001) and number of sprints (*r* = 0.56 [0.18, 0.79]; *p* = 0.004).

Muscle soreness presented *very large* positive correlations with s-RPE (*r* = 0.79 [0.56, 0.91]; *p* < 0.001), exposure time (*r* = 0.83 [0.62, 0.93]; *p* < 0.001), total distance (*r* = 0.81 [0.59, 0.92]; *p* < 0.001) and high-speed distance (*r* = 0.75 [0.47, 0.89]; *p* < 0.001). A *large* positive correlation was found between muscle soreness and number of sprints (*r* = 0.59 [0.22, 0.81]; *p* = 0.002).

Stress presented *unclear* correlations with s-RPE, exposure time, total distance, high-speed distance, and number of sprints (*p* < 0.05).

## 4. Discussion

The purpose of this study was to quantify within-subjects correlations between training load metrics and perceived well-being indices in elite male beach soccer players for three consecutive days of training during the preparation camp for the FIFA Beach Soccer World Cup Russia 2021. Workload metrics (i.e., s-RPE and GPS-derived variables) presented positive associations with perceived well-being indices (i.e., sleep quality, fatigue, and muscle soreness; Hooper Index), meaning the higher the training load metrics, the higher the perceived indices of sleep quality, fatigue, muscle soreness, and Hooper Index (the summation of these ratings). In the present case study, the most intense training day (day 1) had the highest training volume compared with days two and three. This suggests that training volume and intensity might modulate perceived well-being responses, at least under the present conditions.

To our knowledge, this is the first study that quantified within-subjects correlations between workload metrics and perceived well-being indices in elite male beach soccer players. Thus, it is challenging to find some consistent findings in the literature to compare with the results presented here. However, the findings observed in the present investigation were consistent with previous studies [7,12] conducted in male soccer players. For instance, Moalla et al. [12] studied the relationship between daily training load and perceived well-being characteristics of male professional soccer players, reporting that the perceived sleep is moderately related to the daily training load.

In this sense, controlling the training load of beach soccer players (e.g., GPS-derived variables) may also be essential to guarantee a short-term performance development [3]. Moreover, the suitable monitorization of training load will also be important to develop fundamental movement skills, optimize the athlete’s performance, and diminish the injury rate [3]. Therefore, there appears to be no doubt regarding the pertinence of controlling workload metrics to monitor changes in the fitness levels of players [19], to monitor changes in players’ well-being status [7], and to determine the optimal training load session to ensure sufficient recovery and prevent injury [2].

The main findings of the present work, which also corroborates the finding of Nédélec et al. [20], suggest that workloads may affect perceived fatigue, muscle soreness and sleep parameters although further deeper physiological analyses of sleep, fatigue and muscle soreness are required. In fact, perceived well-being indices may be affected not only by workloads but also by other contextual aspects beyond training load, particularly those related to environmental conditions such as high-altitude, religious practices, league ranking and match importance [21]. In addition, and consistent with previous studies [9,22], our findings show that the individual wellness measures are related to training load monitored by the s-RPE based method. In fact, Buchheit et al. [22] demonstrated that perceived ratings of wellness are sensitive to subtle changes in daily training load in elite Australian Rules players. Similarly, Thorpe et al. [9] reported a significant relationship between daily training load and perceived fatigue during the in-season competitive phase in elite soccer players. It is important to point out that the data from Thorpe et al. [9] were collected during the in-season competitive phase in which muscle soreness and sleep quality are well managed to avoid fatigue and overtraining. To the best of our knowledge, the current study is the first to demonstrate significant within-subjects correlations between training load metrics and perceived well-being indices in elite male beach soccer players for three consecutive days of training during the preparation camp for the FIFA Beach Soccer. Many factors such as frequency of administration, time taken to complete the questions, sensitivity of the questionnaire, type of response required, time of day of completion and the amount of time required for appropriate feedback should all be considered in any questionnaire that is implemented [23]. However, we are conscious that the Hooper self-analysis questionnaire is one of the most cost-effective and practical strategies for daily measurements of training effect and early detection of overtraining [24] with a smaller number of items (i.e., four items). The results of this study emphasized the efficacy of the Hooper questionnaire as a simple, practical, useful and non-invasive assessment tool for the daily monitoring of player status and therefore enhancing or at least maintaining the physical performance of soccer players.

It is also important to note that the within-subjects analysis used in the current study may give a more accurate representation of the relationship between workload metrics and perceived well-being indices in elite male beach soccer players. A within-subjects correlation was used to analyze intra-individual association for paired repeated measures, by modeling the longitudinal data set using the correct degrees of freedom [18]. For instance, rather than pooling all the data, or calculating correlations separately for individual subjects, this approach quantifies the correlation, and associated 95% confidence interval, between a covariate and outcome while taking into account the within-subject nature of the study design [18]. Thus, within-subjects correlation has greater statistical power than a simple correlation, and it can better detect associations, as well as prevent spurious correlations [18].

Some limitations should be considered. First, the number of players included was rather low. Secondly, only one team and one training camp over three consecutive days were considered in the study, limiting the generalizability of the findings. Nevertheless, this issue is a common limitation of longitudinal studies in real-world elite sports settings. Despite the relationships (i.e., *moderate* to *very large*) shown between workloads and perceived ratings of wellness in the current study, there are unfortunately no theoretical rationales supporting these findings and the literature is lacking of such investigations, especially in elite beach soccer players. In fact, the first question that we need to solve is how and why workloads influence wellness and vice versa. However, the causal relationship between workloads and wellness measures remains controversial [6], and further research is needed to establish this causal relationship. The lack of additional factors confounding psychometric players’ status could also be considered a limitation of the present study. In fact, wellness parameters could also be influenced by the training schedules and time, strategies of recovery, day of the week and match results. Likewise, the lack of additional biological and physiological measurements is another limitation of the current investigation. Such factors should be taken into consideration in future studies.

For further research, it would be interesting to replicate the present study with more teams, considering the different phases of the season, different levels of competition, or even with other age groups.

## 5. Conclusions

Overall, this case study highlights that workload metrics (i.e., s-RPE and GPS-derived variables) present meaningful within-subjects correlations with perceived well-being indices (i.e., sleep quality, fatigue and muscle soreness; Hooper Index) in elite male beach soccer players during three days of a preparation camp for the FIFA Beach Soccer World Cup Russia 2021. These findings suggest that workload metrics and perceived well-being indices can be implemented in the daily routine of an elite beach soccer team, which may assist coaches, sports scientists, and practitioners in better preparing players for beach soccer competitions.

## Figures and Tables

**Figure 1 ijerph-19-06209-f001:**
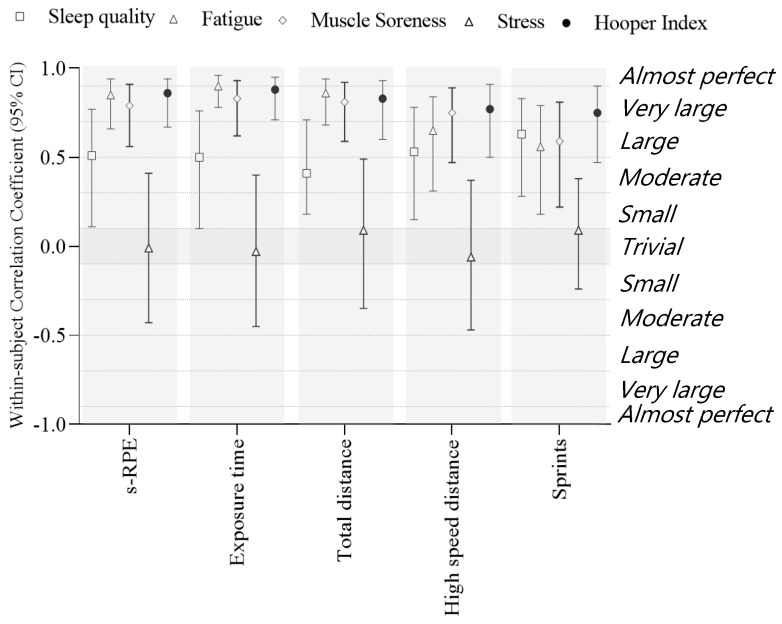
Within-subject correlation coefficients (95% confidence limits) between perceived well-being indices (i.e., sleep quality, fatigue, muscle soreness and stress; Hooper Index) and measures of workload (i.e., s-RPE, exposure time, total distance, high-speed distance, and number of sprints). Correlation coefficient is significant at *p* < 0.05. Abbreviations: s-RPE, session-rating of perceived exertion.

**Table 1 ijerph-19-06209-t001:** Workload metrics and perceived well-being indices in elite male beach soccer players (*n* = 11) during the training camp.

Variables	Day 1	Day 2	Day 3
Sleep quality (a.u.)	3 (1)	2 (1)	1 (1) *^#^
Fatigue (a.u.)	5 (1)	4 (1)	2 (1) *^#^
Stress (a.u.)	3 (2)	2 (1)	2(1)
Muscle soreness (a.u.)	6 (1)	5 (1)	3 (1) *^#^
Hooper Index (a.u.)	18 (4)	12 (2)	8(2) *^#^
s-RPE (a.u.)	465 (399–532)	355 (321–388)	131 (96–165) *^#^
Exposure time (min)	90	75	45 *^#^
Total distance (m)	3006 (2803–3210)	2506 (2376–2637)	1606 (1518–1694) *^#^
High speed distance (m)	154 (136–171)	145 (119–168)	66 (49–84) *^#^
Sprints (*n*)	3 (1–3)	2 (1–2)	0 *^#^

Values are group means and 95% confidence interval (CI) estimates for workload metrics, and group median and interquartile ranges for perceived well-being indices. * Significantly different from day 1 (*p* < 0.05); ^#^ Significantly different from day 2 (*p* < 0.05). Abbreviations: s-RPE, session-rating of perceived exertion; HSD, high-speed distance; a.u., arbitrary units.

## Data Availability

Data cannot be shared publicly because it contains potentially sensitive information and has been obtained from a third party (i.e., Portugal Football School, Portuguese Football Federation) and access restrictions apply. Data are available from the Data Protection Office, Portuguese Football Federation (Data Access contact via e-mail: dpo@fpf.pt) for researchers who meet the criteria to access confidential data.

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
