# Peer review of "Associations between Training Load and Well-Being in Elite Beach Soccer Players: A Case Report"

_ijerph, 2022, doi:10.3390/ijerph19106209_

Round 1
Reviewer 1 Report
First of all, thank you for the opportunity to review this manuscript. The idea of associating well-being and training loads is interesting; however, the study and the manuscript are weak in all the structure:
Abstract:
There is no strong conclusion
Introduction
Too short, without explaining what the authors understand by well-being because the term is broader and more complicated than just quality of sleep, stress, and the others mentioned by the authors. What kind of well-being do the authors measure? Actually, the variable which the authors measure is a physical and mental states. On the other hand, well-being is connected with hedonic and eudaimonic aspects such as positive emotions, engagement, relationships, meaning in life, accomplishments, and more, depending on the well-being model.
The paragraphs are concise, not accurately explain the definitions and the importance of the measured variables.
Material and Methods
The study procedure is simple, making the sample size look very poor, with only eleven beach soccer players.
Results
The sample size is not representative, making the results weak. Only correlations have been made, and why did the authors not make the regression?
Discussion
Very weak and too short
Author Response
Please see in attached file.

Reviewer 2 Report
Abstract
Clear and concise summary of the study
Introduction
Solid introduction that provides the rationale for the study.
Materials
The sample size is appropriate. The data collection is detailed and replicable.
Results
These are detailed and clearly presented. The use of the qualitative description is effective.
Discussion
Well written and provides clear indication of the application of the findings.
Author Response
Please see in attached file.

Reviewer 3 Report
Thank you for the opportunity to review the manuscript entitled "Associations between training load and wellbeing in elite beach soccer players: a case report." This manuscript describes the findings of a repeated measures study of elite male beach soccer players, which assessed the correlation between perceived well-being indices and external training load metrics during the preparation camp for the World Cup.
Congratulations to the authors for an excellent investigation, and working with elite athletes is not an easy task. The topic is an important area for the journal audience, providing practical application information to physiologists and conditioning coaches. I thought the article was well written, the methods were robust, and I enjoyed reading it. The use of mixed models and the calculation of the repeated measures correlation coefficient using the `rmcorr` package was the correct choice of the authors to model data in the presence of the correlated error (longitudinal data analysis).
I have no major corrections. Listed below are some minor comments.
- It would be important to present the 95% CI for the correlation results in the abstract session. But I understand if word count limitation is the cause.
- The authors verified if the residual is normally distributed (Q-Q plot) in the LMM analysis? Please describe in the statistical analysis section (what I mean to say, the ordinal data meets the assumptions of the linear parametric test).
- Technically, the Likert scale data are ordinal. I think it would be better to present the values as median and interquartile ranges for the Hooper Index scale data (Table 1).
Author Response
Please see in attached file.
